Suitability of native milkweed (Asclepias) species versus cultivars for supporting monarch butterflies and bees in urban gardens

Baker Adam M. heresadamb@uky.edu 1
Redmond Carl T. 1
Malcolm Stephen B. 2
Potter Daniel A. dapotter@uky.edu 1
1 Department of Entomology, University of Kentucky , Lexington , KY , USA
2 Department of Biological Sciences, Western Michigan University , Kalamazoo , MI , USA
Colla Sheila
Electronic publication date: 2020 Sep 25
Publication date: 2020
Volume: 8
Electronic Location ID: e9823
Received 2020 Jun 10; Accepted 2020 Aug 5
Copyright: ©2020 Baker et al.
Copyright year: 2020
Copyright holder: Baker et al.
License: This is an open access article distributed under the terms of the Creative Commons Attribution License, which permits unrestricted use, distribution, reproduction and adaptation in any medium and for any purpose provided that it is properly attributed. For attribution, the original author(s), title, publication source (PeerJ) and either DOI or URL of the article must be cited.
License URL: https://creativecommons.org/licenses/by/4.0/

Keywords: Danaus plexippus, Asclepias, Bees, Cultivation, Conservation, Gardens, Urban

Funding: USDA-NIFA-SCRI 2016-51181-235399 IR4:2015-34383-23710 Horticultural Research Institute, BASF Living Acres Program University of Kentucky Nursery Research Endowment Fund USDA-NIFA Hatch Project 2351587000 Funding was provided by USDA-NIFA-SCRI grant 2016-51181-235399 administered through IR4 Grant 2015-34383-23710, the Horticultural Research Institute, BASF Living Acres Program, University of Kentucky Nursery Research Endowment Fund, and USDA-NIFA Hatch Project no. 2351587000. The funders had no role in study design, data collection and analysis, decision to publish, or preparation of the manuscript.

==============================
Public interest in ecological landscaping and gardening is fueling a robust market for native plants. Most plants available to consumers through the horticulture trade are cultivated forms that have been selected for modified flowers or foliage, compactness, or other ornamental characteristics. Depending on their traits, some native plant cultivars seem to support pollinators, specialist insect folivores, and insect-based vertebrate food webs as effectively as native plant species, whereas others do not. There is particular need for information on whether native cultivars can be as effective as true or “wild-type” native species for supporting specialist native insects of conservation concern. Herein we compared the suitability of native milkweed species and their cultivars for attracting and supporting one such insect, the iconic monarch butterfly (Danaus plexippus L.), as well as native bees in urban pollinator gardens. Wild-type Asclepias incarnata L. (swamp milkweed) and Asclepias tuberosa L. (butterfly milkweed) and three additional cultivars of each that vary in stature, floral display, and foliage color were grown in a replicated common garden experiment at a public arboretum. We monitored the plants for colonization by wild monarchs, assessed their suitability for supporting monarch larvae in greenhouse trials, measured their defensive characteristics (leaf trichome density, latex, and cardenolide levels), and compared the proportionate abundance and diversity of bee families and genera visiting their blooms. Significantly more monarch eggs and larvae were found on A. incarnata than A. tuberosa in both years, but within each milkweed group, cultivars were colonized to the same extent as wild types. Despite some differences in defense allocation, all cultivars were as suitable as wild-type milkweeds in supporting monarch larval growth. Five bee families and 17 genera were represented amongst the 2,436 total bees sampled from blooms of wild-type milkweeds and their cultivars in the replicated gardens. Bee assemblages of A. incarnata were dominated by Apidae (Bombus, Xylocopa spp., and Apis mellifera), whereas A. tuberosa attracted relatively more Halictidae (especially Lasioglossum spp.) and Megachilidae. Proportionate abundance of bee families and genera was generally similar for cultivars and their respective wild types. This study suggests that, at least in small urban gardens, milkweed cultivars can be as suitable as their parental species for supporting monarch butterflies and native bees.

Introduction

Burgeoning interest in ecological landscaping to support pollinators, birds, and other urban wildlife is fueling an enthusiastic and active plant movement (Kendle & Rose, 2000; Tallamy, 2008; Jones, 2019; USFS, 2020; USFWS, 2020) and a robust market for native plant species in the nursery, landscape, and gardening trades (Hanson, 2017; ASLA, 2018; Curry, 2018). Native plants can be defined as those that share an evolutionary history with regional insects and other organisms, whereas non-native or exotic plants evolved someplace other than where they have been introduced (Wilde, Gandhi & Colson, 2015). A compelling ecological argument for prioritizing the locally native flora over otherwise desirable (e.g., non-invasive) exotic species is its greater capacity to support local biodiversity, particularly of co-adapted native insect herbivores that are critical food for higher-order consumers including the many species of terrestrial birds that rear their young partly or wholly on insects (Tallamy & Shropshire, 2009; Burghardt, Tallamy & Shriver, 2009; Narango, Tallamy & Marra, 2018). Native plants also support numerous species of pollen-specialist native bees (Fowler, 2016).

Besides promoting plants of local provenance, the horticultural industry has introduced many native plant cultivars, natural variants of native species that are deliberately collected, selected, cross-bred, or hybridized for desirable traits; e.g., disease resistance, plant stature, leaf color, floral display, or extended bloom period, that can be maintained through propagation (Wilde, Gandhi & Colson, 2015). Although use of cultivars is generally discouraged in ecological restoration projects (Lesica & Allendorf, 1999; Kettenring et al., 2014), they are attractive to consumers seeking novel plants that combine the attributes of natives and ornamentals, and open the door to new introductions and vast market potential (Hanson, 2017; Curry, 2018). Indeed, a survey of nurseries in the Mid-Atlantic region, probably representative of the industry overall, found that only 23% of native plant taxa being marketed are true or “wild type”, the rest being available only as cultivated forms (Coombs & Gilchrist, 2017).

Native plant cultivars are not without controversy, however, even for managed landscapes and gardens. Some environmental organizations decry them, arguing that their mass-marketing and use will diminish the genetic diversity of flora in urban ecosystems that are already degraded by preponderance of exotic ornamental plants, further reducing their capacity to adapt to change, support wildlife, or provide other ecosystem services (Wild Ones, 2013). Cultivar traits that could potentially affect pollinator visitation include conversion of anthers and pistils to petals (“double flowered”), color, size, and shape of flowers, floral density, and possibly plant stature (Comba et al., 1999; Corbet et al., 2001; Ricker, Lubell & Brand, 2019). While some floral traits that humans may find attractive in native cultivars, e.g., double flowers or an unusual color, may decrease the quantity, quality, and accessibility of nectar and pollen, making those plants unattractive or of little value to pollinators (Comba et al., 1999; Garbuzov, Alton & Ratnieks, 2017; Mach & Potter, 2018), other native plant cultivars, and many non-natives, do provide high-quality nectar and pollen and can be equally or more attractive to pollinators as native plant species (Masierowska, 2006; Salisbury et al., 2015; White, 2016; Mach & Potter, 2018; Ricker, Lubell & Brand, 2019). Thus, the value of native cultivars for pollinators must be evaluated on a case-by-case basis (Ricker, Lubell & Brand, 2019).

Compared to studies focused on pollinators, little work has addressed the question of whether native plant cultivars are the ecological equivalent to their parent species in supporting native insect folivores. Breeding for traits that change a plant’s form, foliage color, floral display, or phytochemistry could alter cues used by specialist insects in host recognition or acceptance, perhaps to the extent that the insect no longer recognizes or accepts the cultivar as food (Baisden et al., 2018). Alternatively, because there may be tradeoffs in plants’ allocation of resources to defense or growth, selection for traits such as enhanced floral display may make cultivars more palatable to herbivores by reducing their investment in defenses (Herms & Mattson, 1992). Limited research to date suggests the extent to which that may happen depends on the herbivore in question and the particular characteristics of the cultivar that distinguish it from the parent species (Wilde, Gandhi & Colson, 2015). Some cultivar traits, e.g., leaf variegation or leaves altered from green to red or purple, seem to change host suitability for some insects, whereas selection for other traits seems to make little difference insofar as host use by particular herbivores or biodiversity of folivorous insects supported by those plants (Tencazar & Krischik, 2007; Baisden et al., 2018; Poythress & Affolter, 2018). There is particular need for information on whether cultivars of native plants can be as effective as their parental species for supporting specialist native folivores of conservation concern.

The monarch butterfly (Danaus plexippus L.) is arguably the most well-known and beloved native North American insect (Gustafsson et al., 2015). Every fall, hundreds of millions of monarch butterflies make their long-distance journey south from the United States and Canada to overwintering sites in Mexico and California. Both the eastern and western monarch populations declining (Brower et al., 2012; Malcolm, 2018; Rendón-Salinas, Fajardo-Arroyo & Tavera-Alonso, 2015; Pelton et al., 2019) fueling concern that it may face extirpation unless habitat conservation are enacted across North America. Planting milkweeds (Asclepias spp.), the monarch’s obligate larval host plants, is a key part of the international conservation strategy to return this iconic butterfly to sustainable status (Thogmartin et al., 2017; Monarch Joint Venture, 2020; USFWS, 2020). Restoring sufficient milkweed to ensure a stable monarch population will likely require contributions from all land use sectors including urban and suburban areas (Thogmartin et al., 2017; Johnston et al., 2019). In cities and towns, many initiatives are underway, with myriad gardens being planted in residential, educational, and recreational properties (Phillips, 2019; MonarchWatch, 2020; National Pollinator Garden Network, 2020). Milkweed flowers produce abundant nectar and, in addition to monarchs, are highly attractive to bees and numerous other native insects including butterflies, moths, skippers, beetles, and flies (Robertson, 1891; Macior, 1965; Borders & Lee-Mäder, 2015; Baker & Potter, 2018) so urban butterfly gardens can also play a role in supporting their biodiversity. Conservation gardens also provide urban citizens with the opportunity to reconnect with the natural world, helping to foster a greater awareness of conservation issues (Goddard, Dougill & Benton, 2010; Lepczyk et al., 2017; Bellamy et al., 2017).

Native plant cultivars, including milkweeds selected for novel floral display, longer blooming duration, compact growth form, and other consumer-attractive traits, are increasingly available in the wholesale nursery trade and at local garden centers (Baumle, 2018) so it is important to determine if such plants have equivalent value as native species if used for ecological gardening. Different species of milkweeds present a spectrum of palatability across the monarch’s host range (Erickson, 1973; Schroeder, 1976; Baker & Potter, 2018). Milkweed cultivars within a single parental species group may offer a similar spectrum. In this study, we used the high-profile system of milkweeds, monarch butterflies, and bees to test the hypothesis that commercial cultivars provide equivalent ecological benefits as wild-type milkweeds in the context of small urban gardens.

Materials & Methods

Garden study site

Six replicated gardens (1.22 × 9.75 m) were established in public areas of the Arboretum State Botanical Garden of Kentucky, Lexington, in May 2018. The surrounding landscape consisted of restored prairie, formal gardens, and trees. Patches of open, low-maintenance grassland were sprayed with glyphosate to kill existing vegetation, tilled, and covered with weed barrier cloth. Each garden was subdivided into eight randomized 1.22 × 1.22 m plots, one for each of eight milkweed types which included Asclepias incarnata L. (swamp milkweed) and Asclepias tuberosa L. (butterfly milkweed) grown from seedlings produced from commercial open-pollinated seed production fields and hereafter called “wild type” for convenience, and three additional cultivars of each species including A. incarnata ‘Cinderella’, ‘Ice Ballet’, and ‘Soulmate’, and A. tuberosa ‘Blonde Bombshell’, ‘Gay butterflies’ and ‘Hello Yellow’, produced via controlled pollination or tissue culture (Fig. 1, Table S1). The milkweeds were purchased from various producers (American Meadows, Shelburne, VT; Centerton Nurseries, Bridgeton, NJ; Prairie Moon, Winona, MN) as bare root 2-year old plants which were started in our greenhouse. Four plants of a single type (16–30 cm height, depending on species and cultivar) were transplanted 0.6 m apart within each plot. Each garden was then covered with dark brown hardwood mulch (five cm depth). Plants were watered twice per week for the first three weeks to aid establishment and during a period of drought in 2019. We replaced a few of the less-vigorous milkweeds with healthier greenhouse-grown transplants in May 2019 at the start of the second growing season.

Figure 1 Native wild-type milkweed and cultivars as they appeared in the field in 2019.

(A–D) Asclepias incarnata: (A) Wild Type, (B) ’Cinderella’, (C) ’Ice Ballet’, (D) ’Soulmate’. (E–H) Ascelpias tuberosa: (E) Wild Type, (F) ’Blonde Bombshell’, (G) ’Gay Butterflies’, (H) ’Hello Yellow’.

Monarch colonization of wild-type milkweeds and cultivars in gardens

Milkweeds in each garden were monitored for monarch eggs and larvae twice monthly from June to September 2018 and May to August 2019. At each visit all plants were inspected by turning over all leaves, and also examining all stems and flowering portions of the plant. All observations took place between 1000 to 1400 h, on clear warm days. Eggs and larvae were left in place after counting.

Physical and defensive characteristics of wild-type milkweeds and cultivars

Bloom period was assessed in the field for each milkweed type. Plant height and canopy width were measured after bloom when plants had reached maturity. Six leaves (2 each from the upper, middle, and lower thirds of the plant canopy, per milkweed type) were collected from each garden in July 2018, frozen at −80°C, and lyophilized. Cardenolide analysis followed methods of Wiegrebe & Wichtl (1993) and Malcolm & Zalucki (1996).

Trichome densities and latex exudation were compared among milkweeds by methods of Agrawal & Fishbein (2006). Four upper canopy leaves from each replicate (24 total per plant type) were collected in June 2019, leaf discs (28 mm2) were cut about two cm from their tips, and trichomes on adaxial and abaxial surfaces were counted under a binocular microscope. Latex exudation was sampled in the field by cutting the tips (0.5 cm) off intact leaves (24 total per plant type), collecting the exuding latex into pre-weighed tubes with a filter paper wick, and weighing the samples on a microbalance.

Monarch larval performance on wild-type milkweeds and cultivars

Growth and survival of monarch larvae was tested in the greenhouse in July 2019. This trial included two year-old rootstock of the same milkweed species and cultivars in the gardens except for A. tuberosa ‘Blonde Bombshell’ which was excluded because of poor regeneration and market unavailability. All plants were grown in 5.6 liter pots, using a soil and bark mix (SunGro, Quincy, MI), and were 30–60 cm tall. Temperature was regulated between 20−27 ° C and no artificial light was used. Cohorts of newly-molted second instars from our greenhouse colony were placed on plants (one per plant; 10 replicates each) and confined by placing a white fine-mesh bag (25 ×  40 cm) over each plant. Larvae were initially within 12 h of molting, and blocked by slight variation in initial size when allocated to replicates. Potential positional bias was minimized by rotating the position of the plants on the greenhouse bench within each replicate once per day. Larvae were left in place for 7d and then evaluated for amount of weight gained and larval instar level attained.

Bee assemblages of wild-type milkweeds and cultivars

We collected samples of 50 or more bees from blooms of each milkweed type in at least four and in most cases all six of the replicated gardens. Because of sparse blooming of certain milkweed types (mainly A. tuberosa straight species and ‘Hello Yellow’) in one or two of the plots, it was not possible to collect a full sample from every garden. Bees were collected by knocking them into plastic containers containing 70% EtOH, or sometimes caught with aerial nets held over an umbel so that bees would fly up into the net, or by gently sweeping the blooms without damaging the plant. We collected the first 50 bees encountered on a given milkweed species per replicate which required multiple visits to each garden during peak bloom. At each visit, we placed eight bee collection containers (one for each milkweed type) in each garden, and then worked our way through all replicates, starting at a different garden on each visit, collecting bees systematically throughout. Bee samples were washed with water and dish soap, rinsed, then dried using a fan–powered dryer for 30–60 min and pinned. Specimens were identified to genus (Packer, Genaro & Sheffield, 2007), with honey bees and bumble bees taken to species (Williams et al., 2014).

Data analyses

We used separate two-way analyses of variance (ANOVA) for a randomized complete block design to compare numbers of monarch eggs and larvae in gardens, larval performance, and plant characteristics between all milkweed types, and within milkweed species. Two-tailed Dunnett’s tests were used when the F- statistic was significant to test for differences among individual cultivars and their parental milkweed species.

Bee genus richness and diversity (Simpson Index of Diversity 1-D; Magurran, 2004) were similarly compared. Statistical analyses were performed with Statistix 10 (Analytical Software, 2013). Chi-square analyses were used to compare proportionate representation of bee families in samples from the wild type and cultivars within each milkweed species. Data are reported as means ±  standard error (SE).

Results

Monarch colonization of wild-type milkweeds and cultivars in gardens

Each of the six gardens attracted monarchs, with eggs and larvae found throughout the 2018 and 2019 growing seasons (238 and 207 total individuals, respectively). Monarch immature life stages were first found in the gardens in May, peaking in August and persisting into September. Significantly more eggs and larvae were found on A. incarnata than A. tuberosa in 2018 (F7,47 = 5.25, P <  0.001) and 2019 (F6,41 = 6.29, P <  0.001) but within species, there were no differences in extent of colonization of the wild types versus their cultivars in either year (Table 1). The A. tuberosa cultivar ‘Blonde Bombshell’ was excluded in 2019 due to poor regeneration of the in-ground plants and market unavailability for replacements.

Table 1 Monarch colonization of wild-type milkweed and cultivars in replicated outdoor gardens, and larval performance on those milkweeds in the greenhouse, showing within-species similarity.

	Eggs and larvae on milkweeds in gardensa		Larval performance after 7 d on plants in the greenhouseb	
Species and cultivar	2018	2019			Weight (mg) attainedc	Instar attainedd	No. live (of 10)	
A. incarnata								
Wild type	7.7 ± 2.6	7.3 ± 1.0			436 ± 81	3.7 ± 0.2	9	
‘Cinderella’	11.7 ± 3.4	11.0 ± 1.6			392 ± 43	3.5 ± 0.2	10	
‘Ice Ballet’	7.7 ± 3.0	13.3 ± 3.5			417 ± 42	3.3 ± 0.2	9	
‘Soulmate’	8.7 ± 2.4	12.3 ± 2.4			386 ± 59	3.6 ± 0.3	9	
F3,15	0.80	1.08		F3,24	0.14	0.52		
P	0.51	0.39		P	0.94	0.67		
A. tuberosa								
Wild type	1.7 ± 0.8	2.2 ± 0.6			1122 ± 184	4.6 ± 0.2	9	
‘Blonde Bombshell’e	0.5 ± 0.3	–			–	–		
‘Gay Butterflies’	1.8 ± 0.6	3.3 ± 1.8			1175 ± 155	4.8 ± 0.2	8	
‘Hello Yellow’	1.2 ± 0.6	2.2 ± 0.9			739 ± 70	4.3 ± 0.2	10	
F3,15[2,10]	1.33	0.35		F2,15	3.2	1.55		
P	0.30	0.71		P	0.07	0.24		
Notes.

Data are means ± SE for each milkweed type.

a Eggs and larvae were more abundant on A. incarnata than A. tuberosa in 2018 (F7,47 = 5.25, P <0.001) and in 2019 (F6,41 = 6.29, P <0.001).

b Newly-molted second instars (n = 10) were reared individually on separate plants.

c Larval weight differed significantly among milkweed types (F 6,48 = 12.42; P <0.001) and was greater on A. tuberosa, as a group, than on A. incarnata (contrastst = 8.1; P<0.001).

d Larval instar differed significantly among milkweed types (F 6,48 = 7.95; P<0.001) and was greater on A. tuberosa, as a group, than on A. incarnata (contrasts; t = 6.65; P <0.001).

e Blonde Bombshell was excluded in 2019 due to poor regeneration in the gardens and market unavailability for the greenhouse trial.

Defensive and physical characteristics of wild-type milkweeds and cultivars

Expression of defensive characteristics differed among milkweed types (Table 2). There was no overall significant difference in latex expression between the two milkweed species, but A. tuberosa, as a group, had relatively more trichomes and higher cardenolide concentrations (Table 2). Within the A. incarnata group, ‘Cinderella’ had significantly higher latex expression than the wild type, and ‘Ice Ballet’ had the highest number of trichomes and highest cardenolide concentrations. Within the A. tuberosa group ‘Gay Butterflies’ and ‘Hello Yellow’ had significantly higher latex expression than the wild type.

Table 2 Defensive characteristics of native wild-type milkweeds and cultivars.

	Latex (mg exuded)a	Trichomes per 28 mm2	Cardenolides (μ g/g)	
A. incarnata				
Wild Type	1.4 ± 0.2	97 ± 13	4.6 ± 1.8	
‘Cinderella’	3.4 ± 0.8*	93 ± 14	4.9 ± 2.8	
‘Ice Ballet’	1.1 ± 0.2	131 ± 13*	18.5 ± 6.3*	
‘Soulmate’	1.1 ± 0.2	92 ± 14	12.2 ± 3.4	
	F3,35 = 11.2	F3,67 = 3.1	F3,15 = 2.3	
	P <0.001	P = 0.03	P = 0.01	
A. tuberosa				
Wild Type	0.7 ± 0.2	212 ± 17	392 ± 93	
‘Blonde Bombshell’	–	–	489 ± 148	
‘Gay Butterflies’	2.1 ± 0.4*	202 ± 27	684 ± 535	
‘Hello Yellow’	2.3 ± 0.3*	153 ± 21	498 ± 296	
	F2,31 = 14.4	F2,64 = 2.6	F3,14 = 0.25	
	P <  0.001	P = 0.08	P = 0.86	
Notes.

Data are means ± SE for each milkweed type.

a amount exuded from cut leaves (n = 24 per plant type, 4 per garden).

* denotes significant within-species difference from straight species by 2-tailed Dunnett’s test (P <0.05).

Asclepias incarnata, as expected, were taller than A. tuberosa (Table S2). Plant stature was similar within the A. incarnata group except for cultivar ‘Soulmate’ which had a wider canopy than the wild type. Within A. tuberosa, ‘Gay Butterflies’ and ‘Hello Yellow’ were taller and wider than the wild type. All of the milkweeds bloomed in June and July.

Larval performance on of wild-type milkweeds and cultivars

Monarch larvae grew and developed on all milkweeds tested (Table 1). Growth and development were faster overall on A. tuberosa than on A. incarnata, but within groups was similar on wild types and their respective cultivars.

Bee assemblages of garden milkweeds

Five families and 17 genera were represented amongst the total of 2436 bees sampled from milkweed blooms in the replicated garden plots (Table 3). Within the A. incarnata group, bee genus diversity was similar (F3,15 = 1.74, P = 0.2) but genus richness was greater for ‘Soulmate’ than for the wild type (F3,15 = 4.14, P = 0.03; Table 3). Bee genus diversity was similar within the A. incarnata group ( F3,15 = 1.74, P = 0.2, Table 3). Bee assemblages of A. incarnata were dominated by Apid bees (Fig. 2), particularly bumble bees (Bombus spp.), carpenter bees (Xylocopa spp.), and honey bees (Apis mellifera). Representation of particular families and genera was similar among the four types except for ‘Soulmate’ which attracted proportionately few Bombus spp. compared to the wild type (χ2 = 29.5, P <  0.001).

Table 3 Bee assemblages of two species of native milkweeds and their cultivars in replicated gardens based on total collected across all sampling dates.

	A. incarnata and cultivarsa	A. tuberosa and cultivarsb	
	WT	CN	IB	SM		WT	BB	GB	HY	
Andrenidae										
Andrena sp.	0	1	0	0		0	0	0	0	
Apidae										
Apis mellifera	16	60	47	52		27	31	79	29	
Bombus bimaculatus	0	12	0	2		6	1	5	9	
B. griseocollis	137	213	165	110		41	9	117	75	
B. impatiens	0	1	5	0		4	3	29	16	
B. pensylvanicus	0	0	0	1		0	0	0	0	
Ceratina sp.	0	0	0	0		2	0	11	4	
Xylocopa virginica	82	80	32	104		5	0	5	3	
Colletidae										
Hylaeus sp.	2	3	2	14		0	6	1	0	
Halictidae										
Agapostemon sp.	0	2	1	1		2	1	8	1	
Augochlora sp.	1	0	0	1		10	11	16	4	
Augochlorella sp.	0	4	0	6		1	5	15	1	
Augochloropsis sp.	1	6	9	15		8	3	7	5	
Halictus sp.	0	2	3	0		5	15	5	0	
Lasioglossum sp.	11	20	24	39		83	224	91	45	
Sphecodes sp.	0	0	1	0		0	0	0	0	
Megachilidae										
Anthidium sp.	0	0	0	0		4	0	2	2	
Coelioxys sp.	0	0	0	1		10	1	10	3	
Heriades sp.	0	0	0	0		4	0	0	1	
Megachile sp.	0	0	3	3		14	6	35	6	
Total bees sampled	250	404	291	346		227	317	398	203	
Mean genus richness(SE)	4.5 (0.2)	5.3 (0.6)	6.2 (0.7)	7.0*(0.5)		7.7 (1.0)	6.3 (1.0)	10.0 (0.6)	6.5 (0.4)	
Mean genus diversityc (SE)	0.59 (0.04)	0.61 (0.08)	0.63 (0.03)	0.74 (0.04)		0.74 (0.11)	0.46 (0.07)	0.75 (0.02)	0.83 (0.02)	
Notes.

a WT, Wild Type; CN, ‘Cinderella’; IB, ‘Ice Ballet’; SM, ‘Soulmate’.

b WT, Wild Type; BB, ‘Blonde Bombshell’, GB, ‘Gay Butterflies’, HY, ‘Hello Yellow’

c Simpson Index of Diversity 1-D (Magurran, 20014) calculated across all six gardens.

* Cultivar mean differs significantly from mean for wild type (Dunnett’s test, P < 0.05). See text for ANOVA results for genus richness and diversity.

Asclepias tuberosa attracted a somewhat more even distribution of bee families and genera, with proportionately more Halictidae and Megachilidae compared to the A. incarnata group, and each cultivar attracting diverse bee genera in varying proportions (Table 3, Fig. 3). Although A. tuberosa ‘Blonde Bombshell’ attracted bees from 11 different genera, most (71%) of them were Halictidae, genus Lasioglossum, accounting for that cultivar having lower genus diversity than the wild type (F3,15 = 5.82, P = 0.007). There was significant variation in bee genus richness within the A. tuberosa group (F3,15 = 6.31, P <  0.01) but none of the cultivars had higher or lower richness than did the wild type (Table 3).

Figure 2 Relative proportions of bee families (A–D) and genera (E–H) collected from A. incarnata wild type and cultivars. (A,E) Wild Type; (B,F) ’Cinderella’; (CG) ’Ice Ballet’; (DH) ’Soulmate’.

Figure 3 Relative proportions of bee families (A–D) and genera (E–H) collected from A. tuberosa wild type and cultivars. (A,E) Wild type; (B,F) ’Blonde Bombshell’; (C,G) ’Gay Butterflies’; (DH) ’Hello Yellow’.

Discussion

A major challenge to scaling up the use of native species in landscaping and gardening is providing plants that are both ecologically functional and profitable for the horticulture industry (Wilde, Gandhi & Colson, 2015). Native plants are mainly introduced into urban ecosystems through a market system that satisfies consumer preferences for ornamental traits. Consequently, many native plant species have been selected or bred for extended flowering, novel color, size, or morphology of flowers or foliage, compactness, or other aesthetic characteristics, with frequent new cultivar introductions (Wilde, Gandhi & Colson, 2015). Depending on their traits, some native plant cultivars seem to support specific folivorous insects, or insect-based food webs, as effectively as native plant species, whereas others do not (e.g., Tencazar & Krischik, 2007; Baisden et al., 2018; Poythress & Affolter, 2018; Ricker, Lubell & Brand, 2019). There is particular need for information on whether or not cultivars can support native insects of conservation concern.

Among such insects, none approaches the power of the monarch butterfly as a catalyst for public interest in ecological gardening (Gustafsson et al., 2015). Our results suggest that, at least in urban pollinator gardens, cultivars of A. incarnata and A. tuberosa, two of the most widely-sold garden-friendly native milkweeds (Baker & Potter, 2018), are as suitable as their respective parental species for attracting and supporting monarch butterflies. Over two growing seasons, we found similar numbers of naturally-occurring eggs and larvae on cultivars and straight species within each group. Despite some differences in plant defensive characteristics (trichomes, latex, and cardenolides), larval growth, development, and survival were similar on milkweeds within each group. Monarch larvae are capable of dealing with a range of milkweed defenses (Dussourd & Eisner, 1987; Agrawal & Fishbein, 2006). It is not unexpected, therefore, that cultivation at least within A.incarnata and A. tuberosa does not result in changes in defense that are too severe for monarch larvae to overcome.

Shared evolutionary history with plants has led to widespread host specificity in phytophagous insects (Bernays & Graham, 1988). Many Lepidoptera have narrow host ranges, often restricted to a single genus (Dyer et al., 2007), so a plant breeder selecting for modified plant phenotypes could potentially alter the cues such insect specialists rely upon to recognize their hosts. Butterflies, in general, use a combination of visual, olfactory, and gustatory cues to find and accept host plants (Renwick & Chew, 1994). Monarchs move extensively between habitat patches, but the relative distance over which they use vision or olfaction to locate milkweeds or nectar sources is uncertain (Zalucki, Parry & Zalucki, 2016).

Monarch females foraging in natural habitat tend to lay more eggs on taller, more isolated milkweed plants than on shorter, less accessible ones (Zalucki & Kitching, 1982; Zalucki, Parry & Zalucki, 2016), and the same patterns occur in butterfly gardens (Baker & Potter, 2018; Baker & Potter, 2019). The relatively short stature of all cultivars of A. tuberosa (Table S2) compared to A. incarnata may account, in part, for why we found fewer eggs and larvae on the former species in both years despite them both being suitable as larval food (Erickson, 1973). Shorter milkweeds may go unnoticed by the butterflies because they are less visually apparent and accessible than taller milkweeds, especially when surrounded by non-host plants (Baker & Potter, 2019).

Some other butterfly species form a visual search image for host plants with a particular leaf shape that facilitates host-finding in the field (Benson, Brown & Gilbert, 1975; Rausher, 1978; Dell’Aglio, Lasada & Jiggins, 2016), but it is not known if monarchs do this. The estimated 100 milkweed species native to North America vary in leaf size and shape (Woodson Jr, 1954), and several studies suggest that those with narrow leaves (e.g., A. verticillata) are less preferred for oviposition (Baker & Potter, 2018; Pocius et al., 2018). All native cultivars used in our study had leaves seemingly similar to their parental species, but if plant breeders were to select for cultivars having modified leaf shape, color, or variegation, such changes could potentially affect monarchs’ visual perception of them as hosts.

Native bee populations are declining (Cameron et al., 2011; Koh et al., 2016) and millions of urban pollinator gardens are being planted to help their plight (Phillips, 2019). Milkweed flowers produce abundant nectar (Wyatt & Broyles, 1994), and are highly attractive to bees and other nectar-feeding insects (Fishbein & Venable, 1996; MacIvor et al., 2017; Baker & Potter, 2018). Because milkweed pollen is enclosed within pollinia, nectar is the only reward that milkweeds offer their pollinators (Wyatt & Broyles, 1994). Large bees and wasps are the most effective milkweed pollinators, whereas most of the smaller visitors are unable to transfer pollina and do not provide pollination services to milkweed (Kephart, 1983; Ivey, Martinez & Wyatt, 2003; MacIvor et al., 2017).

In the present study, large-bodied, eusocial Apidae dominated the bee assemblages of A. incarnata whereas A. tuberosa attracted proportionately more Halictidae, Megachilidae, and other relatively small native bees. Both patterns are consistent with an earlier study in which only wild-type milkweeds were compared (Baker & Potter, 2018). Large apid bees have high energy demands (Heinrich, 1976), so may favor milkweeds such as A. incarnata having large flowers and abundant nectar rewards, whereas the relatively smaller flowers of A. tuberosa may provide a sufficient nectar reward for relatively smaller native bees (Baker & Potter, 2018). Unlike garden plants wherein cultivar selection has reduced or eliminated floral rewards for pollinators (Garbuzov, Alton & Ratnieks, 2017; Erickson et al., 2019), all of the native milkweed cultivars we evaluated were bee-attractive. Moreover, with the possible exception of A. tuberosa ‘Blonde Bombshell’ which attracted an inordinately high number of Lasioglossum sp., bee assemblages of the milkweed cultivars were generally similar to those of their respective parental species.

Conclusions

Restoration ecologists, conservation groups, and U.S. federal and state agencies are promoting increased use of native plants in landscaping and gardening to help support biodiversity in urbanized areas. A major challenge to that goal is availability of native plants that satisfy requirements for ecological function, cost-effective production, and desirable ornamental characteristics with consumer appeal. Breeding, marketing, and use of native plant cultivars is widespread and growing in the horticulture industry. This study suggests that, at least in small gardens, native milkweed cultivars can be as suitable as their parental species for attracting and supporting monarch butterflies and native bees. Although probably not appropriate for use in natural areas where maintaining a reservoir of genetic variability is important for plant population resilience, use of native milkweed cultivars in pollinator gardens can help support the urban public’s contribution to monarch and native bee conservation. For urban gardens, planting several species of native milkweeds, regardless of whether they are wild types or native cultivars, plus a variety other plants to provide nectar and pollen throughout the growing season, is probably the best strategy for helping to support monarchs, bees, and other pollinators.

Supplemental Information

Supplemental Information 1 Ornamental characteristics of wild-type milkweeds and cultivars

Click here for additional data file.

Supplemental Information 2 Height and width at maturity and bloom period of milkweeds in the replicated gardens in 2019

*denotes significant difference compared to wild-type within species, ANOVA, 2-tailed Dunnett’s test, P < 0.001

Click here for additional data file.

Supplemental Information 3 Raw data

Click here for additional data file.

We thank the staff at the Arboretum and State Botanical Garden of Kentucky, particularly Jesse Dahl, for help establishing and maintaining the garden plots, H Cybriwsky, M Geise, K O’Hearn, and B Riordan for help with field sampling, and R Geneve for technical information about cultivars.

Additional Information and Declarations

Competing Interests

Author Contributions

Data Availability

The authors declare there are no competing interests.

Adam M. Baker conceived and designed the experiments, performed the experiments, analyzed the data, prepared figures and/or tables, authored or reviewed drafts of the paper, and approved the final draft.

Carl T. Redmond performed the experiments, prepared figures and/or tables, authored or reviewed drafts of the paper, and approved the final draft.

Stephen B. Malcolm performed the experiments, authored or reviewed drafts of the paper, and approved the final draft.

Daniel A. Potter conceived and designed the experiments, analyzed the data, prepared figures and/or tables, authored or reviewed drafts of the paper, and approved the final draft.

The following information was supplied regarding data availability:

The raw measurements are available in the Supplementary Files.

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
