# Peer review of "Suitability of native milkweed (Asclepias) species versus cultivars for supporting monarch butterflies and bees in urban gardens"

_PeerJ, doi:10.7717/peerj.9823_

## Round 0.1 · original submission · Minor Revisions

Please clearly address the minor revisions provided by the two reviewers. I look forward to your re-submission.

Reviewer 1 ·

Basic reporting

• Happy to see this work. Cultivars, especially nativars, are a big question in the field of pollinator conservation and pollinator-friendly gardening. There has been little work to date on it.
• Raw data was not shared for field and greenhouse counts of larvae, eggs, instar development, or for bee visitors to flowers, so analyses could not be checked; the authors do say in the paper they will share it with others upon request
• Line 403/reference section: Access dates are commonly provided with website references but are not in this paper
• Figures look pixelated (photos, text, and diagrams); if it’s not due to the proof creation itself, you may want to reconsider quality
• Consider the appearance and contrast of the colours of your figures – perhaps add hatching or similar to distinguish; even in colour Megachile & Hylaeus, for instance, look similar, and if anyone were to view these in black & white, the colours could be even harder to distinguish
• Table 3: include information in the title as to what these numbers are; counts of each species of bee over all sampling times and treatment blocks combined? Similarly, explain what diversity index you are using here too.
• Table S1: it is currently hard to tell what text belongs to which species/cultivar; if the format for PeerJ does not allow for horizontal lines, add more space in between the lines (this may be corrected in the final type-set paper but should be checked before publication)

Experimental design

o The paper does describe original research that is relevant and meaningful. I do have some comments and questions, primarily for clarification, although some additional analyses would be welcome.
o Line 135, 148, etc.: why, in addition to monarchs, are you focussing on bees and not other taxa? Discuss.
o Line 156: grassland (i.e. tall grasses, flowers, etc.) or turf? What is in the surrounding landscape?
o Line 163/Table S1: you indicate that the cultivar ‘Gay Butterflies’ can have three different bloom colours; are these different on a single plant, or within a flower? What did the plants in your experiment look like? The colour could affect your results vs that of others in the future/what gardeners would find.
o Table S1: Clarify that controlled vs open pollination means, and where this data came from if not from your observations
o Line 166: started in your own greenhouse after purchase and prior to planting, vs the vendor’s greenhouse?
o Line 171: presumably all observations occurred one the same day for all plots?
o Line 172: what times of the day did you do the investigations? Did you consider weather conditions?
o Line 172ish: did you do any observations on adult monarchs in the area (counts & observations of behaviours)?
• Line 174/Table 1: why are you combining the number of eggs and larvae together? I would be interested in seeing the differences between these numbers. E.g. are more eggs laid but don’t survive on one species/nativar than another?
o Line 198: repetition/duplication of the word adaxial
o Line 199: why cut the tips off vs e.g. the edges of the leaf? Caterpillars do not solely feed on tips.
o Line 209: Where did the instars come from for your greenhouse experiments? How did you confirm that they were all the same age (i.e. is newly molted recognizable and thus all the instars were within a few hours of each other vs a day/days)?
o Line 211: did you change the position of where you put the instars on the plants or you changed the plant locations? If you changed the plants, how were they arranged so that you could move them vertically (i.e. on shelving?)?
o Line 213: suggest adding in ‘stage’ or ‘level’ for instar attained
o Line 216: did you sample the first 50 bees seen then to avoid collection bias? Did you change the order of approach to the species in the beds?
o Line 226: you do not discuss chi-square analyses here but do present the results later, for bee diversity

Validity of the findings

• I believe the findings of the authors are valid, but would like some more clarification on statistics, some additional statistics, and an expanded discussion.
• Line 249/Table 2: you don’t present the Dunnet’s results for the stats, just stated they are significant
• Line 257/Table S2: you don’t present the ANOVA results for the between or within group comparisons
• Line 270: why don’t you compare visitors between the two groups of milkweed? i.e. number of visitors, species, etc? are there significant differences between cultivars (within group) or between groups – you list richness and diversity but don’t statistically compare? Figure 2 clearly shows that there are differences, e.g. Ice Ballet vs Soul mate for incarnata, wild type vs Hello yellow for tuberosa.
• Line 280: do you mean Table 3 not table 1?
• Table 3: shouldn’t genus richness just be the total number of different genera observed per plant? Because the numbers don’t add up that way (e.g. for A. incarnata wild type, 7 genera were observed but the richness is 5). Also, how do you have a mean and se for diversity index per cultivar? Are you doing it by plot or something vs overall?
• Line 297: this is the first time you mention insects of conservation concern; if this is a reason for your research, you should mention it in your intro and perhaps try to identify species of concern that would be relevant besides monarchs
• Line 354: but there were significant differences found in insect visitation, diversity, between cultivars and species. Why is this? Any hypotheses? You reference Baker & Potter 2018 for wild type studies – did they discuss why there is a difference?
• Line 354: discuss bee (& other pollinator) vision and search patterns and how different coloured flowers may effect their visitation rates (esp. white flowers of Ice Ballet vs pink of wild type; maybe no difference seen for bees but would other groups be affected?)
• Did you notice any predation happening of the larvae/eggs? Did this seem to differ between species?
• Why was egg/instar development faster on tuberosa than incarnata?

Additional comments

• Some additional comments, primarily for clarification:
• Line 30: I suggest including a brief description of the nativars used in your abstract (e.g. were they similar in appearance to the natives)
• Line 100-ish: Annie White has also been doing some work in this area; see e.g. her thesis or website (https://pollinatorgardens.org/2013/02/08/my-research/)
o White, Annie, "From Nursery to Nature: Evaluating Native Herbaceous Flowering Plants Versus Native Cultivars for Pollinator Habitat Restoration" (2016). Graduate College Dissertations and Theses. Paper 626.
o Line 119/throughout: be sure to include the taxonomic authority for every species you list, whether it be plant or insect (just need to do this for the first use of the species name)
o Line 135: milkweeds are visited by many insect taxa besides monarchs and bees (e.g. beetles, wasps, moths, other butterflies, flies, other arthropods); I suggest mentioning them or at least acknowledging this point in your introduction
o Line 163/Table S1: why do you use the precise color descriptor of “kelly green” but then less precise color descriptors like light green, dark green?
o Line 163/Table S1: provide further clarification on additional features; i.e. larger flower clusters – larger in number of flowers per cluster, larger in flower size, how much larger?
o Line 260/Table S2: I suggest including actual dates of bloom, and/or number of days/weeks in bloom for each speies, as “June-July” could be 2 weeks or 8 weeks long and you’re just presenting results for one year; and/or discuss period when e.g. 10% bloom, 50% bloom, 75% bloom, 100% bloom
o Line 310: should be a period and not a comma between A. incarnata
o Line 364: add the word “in” in at the end of the line
o Line 542: You may wish to investigate and then replace the website blog reference you used for Keith Nevison with his actual MSc thesis, which is available online at http://udspace.udel.edu/handle/19716/21442
• Discuss general preferred soil conditions for each species, and their general persistence elsewhere & overtime (e.g. incarnata prefers wet soils but can grow in dry, but sometimes not as well over time)

Reviewer 2 ·

Basic reporting

The manuscript is extremely well-written throughout. The overall issue/research question is clearly and concisely presented and the authors provide sufficient background/context about this important issue. The overall manuscript structure is well throughout out and again easy to follow. The associated figures/tables are clear. Thus the manuscript meets all the criteria of this reporting section.

Experimental design

The overall research presented, in my opinion, fits within the journal's Aims and Scope appropriately.

The authors present a clearly defined research question, which is an emerging area of interest, particularly as it pertains to the Green Industry, green marketing, monarch and native pollinator conservation, and urban biodiversity conservation/green networks. It will also have significant value to extension agents, APGA professional, etc. The results evaluating native type vs cultivars indeed fills a needed, identified niche knowledge gap.

I feel that the methods/research design are appropriate, adequately rigorous, and clearly presented (appropriate replication, well-designed, and described such that another researcher could replicate from the information provided- and as such meet the appropriate standards for publication. I also feel that the depth of design is well conceived - evaluating organism attraction, use, plant chemical and mechanical defense, plant performance, etc. All the key variables essential to measure and that are relevant to help address the initial research question.

Validity of the findings

As in # 2 above, I feel that the results are novel and important as they pertain to the Green Industry, green marketing, monarch and native pollinator conservation, and urban biodiversity conservation/green networks. It will also have significant value to extension agents, APGA professional, etc. Such pollinator attraction, plant variety trials need to be expanded to better elucidate key questions for urban biodiversity conservation impact.

Data are sufficient, analyses is appropriate and statistically sound in my opinion.

The overall conclusions are well stated, appropriately linked to the original research question and supported by the results.

Additional comments

Please see my comments and questions on the manuscript PDF itself. Overall, this is a very well crafted study and manuscript that has significant benefits to urban biodiversity/monarch conservation and the Green Industry as a whole. I wonder though if it would be worth mentioning in the conclusion some cautions regarding nursery purchased plants such as the use of systemic insecticides that may have lethal or sublethal impacts to insects. Such may be the case with cultivars offered at larger retailers that often do not carry true native types - with those being more commonly sold from specialty or native plant nurseries.
.

Annotated reviews are not available for download in order to protect the identity of reviewers who chose to remain anonymous.

---

## Round 0.2 · accepted · Accept

Thank you for addressing the reviewer comments.